# Provably Strict Generalisation Benefit for Invariance in Kernel Methods

**Bryn Elesedy**
University of Oxford
`bryn@robots.ox.ac.uk`

## Abstract

It is a commonly held belief that enforcing invariance improves generalisation. Although this approach enjoys widespread popularity, it is only very recently that a rigorous theoretical demonstration of this benefit has been established. In this work we build on the function space perspective of Elesedy and Zaidi [8] to derive a strictly non-zero generalisation benefit of incorporating invariance in kernel ridge regression when the target is invariant to the action of a compact group. We study invariance enforced by feature averaging and find that generalisation is governed by a notion of effective dimension that arises from the interplay between the kernel and the group. In building towards this result, we find that the action of the group induces an orthogonal decomposition of both the reproducing kernel Hilbert space and its kernel, which may be of interest in its own right.

## 1 Introduction

Recently, there has been significant interest in models that are invariant to the action of a group on their inputs. It is believed that engineering models in this way improves sample efficiency and generalisation. Intuitively, if a task has an invariance, then a model that is constructed to be invariant ahead of time should require fewer examples to generalise than one that must learn to be invariant. Indeed, there are many application domains, such as fundamental physics or medical imaging, in which the invariance is known a priori [29, 32]. Although this intuition is certainly not new (e.g. [33]), it has inspired much recent work (for instance, see [36, 15]).

However, while implementations and practical applications abound, until very recently a rigorous theoretical justification for invariance was missing. As pointed out in [8], many prior works such as [28, 24] provide only worst-case guarantees on the performance of invariant algorithms. It follows that these results do not rule out the possibility of modern training algorithms automatically favouring invariant models, irrespective of the choice of architecture. Steps towards a more concrete theory of the benefit of invariance have been taken by [8, 20] and our work is a continuation along the path set by [8].

In this work we provide a precise characterisation of the generalisation benefit of invariance in kernel ridge regression. In contrast to [28, 24], this proves a *provably strict* generalisation benefit for invariant, feature-averaged models. In deriving this result, we provide insights into the structure of reproducing kernel Hilbert spaces in relation to invariant functions that we believe will be useful for analysing invariance in other kernel algorithms.

The use of feature averaging to produce invariant predictors enjoys both theoretical and practical success [17, 9]. For the purposes of this work, feature averaging is defined as training a model as normal (according to any algorithm) and then transforming the learned model to be invariant. This transformation is done by *orbit-averaging*, which means projecting the model on the space of invariant functions using the operator $\mathcal{O}$ introduced in Section 2.3.

Kernel methods have a long been a mainstay of machine learning (see [30, Section 4.7] for a brief historical overview). Kernels can be viewed as mapping the input data into a potentially infinite

35th Conference on Neural Information Processing Systems (NeurIPS 2021).

dimensional feature space, which allows for analytically tractable inference with non-linear predictors. While modern machine learning practice is dominated by neural networks, kernels remain at the core of much of modern theory. The most notable instance of this is the theory surrounding the *neural tangent kernel* [11], which states that the functions realised by an infinitely wide neural network belong to a reproducing kernel Hilbert space (RKHS) with a kernel determined by the network architecture. This relation has led to many results on the theory of optimisation and generalisation of wide neural networks (e.g. [14, 3]). In the same vein, via the NTK, we believe the results of this paper can be extended to study wide, invariant neural networks.

## 1.1 Summary of Contributions

This paper builds towards a precise characterisation of the benefit of incorporating invariance in kernel ridge regression by feature averaging.

Lemma 3, given in Section 3, forms the basis of our work, showing that the action of the group $\mathcal{G}$ on the input space induces an orthogonal decomposition of the RKHS $\mathcal{H}$ as

$$\mathcal{H} = \overline{\mathcal{H}} \oplus \mathcal{H}_\perp$$

where each term is an RKHS and $\overline{\mathcal{H}}$ consists of all of the invariant functions in $\mathcal{H}$. We stress that, while the main results of this paper concern kernel ridge regression, Lemma 3 holds regardless of training algorithm and could be used to explore invariance in other kernel methods.

Our main results are given in Section 4 and we outline them here. We define the generalisation gap $\Delta(f, f')$ for two predictors $f, f'$ as the difference in their test errors. If $\Delta(f, f') > 0$ then $f'$ has *strictly better test performance* than $f$. Theorem 5 describes $\Delta(f, f')$ for $f$ being the solution to kernel ridge regression and $f'$ its invariant (feature averaged) version and shows that it is positive when the target is invariant.

More specifically, let $X \sim \mu$ where $\mu$ is $\mathcal{G}$-invariant and $Y = f^*(X) + \xi$ with $f^*$ $\mathcal{G}$-invariant and $\mathbb{E}[\xi] = 0$, $\mathbb{E}[\xi^2] = \sigma^2 < \infty$. Let $f$ be the solution to kernel ridge regression with kernel $k$ and regularisation parameter $\rho > 0$ on $n$ i.i.d. training examples $\{(X_i, Y_i) \sim (X, Y) : i = 1, \ldots, n\}$ and let $f'$ be its feature averaged version. Our main result, Theorem 5, says that

$$\mathbb{E}[\Delta(f, f')] \geq \frac{\sigma^2 \dim_{\text{eff}}(\mathcal{H}_\perp) + \mathcal{E}}{(\sqrt{n} M_k + \rho/\sqrt{n})^2}$$

where $M_k = \sup_x k(x, x) < \infty$, $\mathcal{E} \geq 0$ describes the approximation errors and $\dim_{\text{eff}}(\mathcal{H}_\perp)$ is the effective dimension of the RKHS $\mathcal{H}_\perp$. For an RKHS $\mathcal{H}$ with kernel $k$ the effective dimension is defined by

$$\dim_{\text{eff}}(\mathcal{H}) = \int_{\mathcal{X}} k(x, y)^2 \, \mathrm{d}\mu(x) \, \mathrm{d}\mu(y).$$

where $\mathcal{X} = \operatorname{supp} \mu$. We return to this quantity at various points in the paper.

It is important to note that the use of the feature averaged predictor $f'$ as a comparator is without loss of generality. Any other predictor $f''$ that has test risk not larger than $f'$ would satisfy the above bound, simply because this means $\Delta(f', f'') \geq 0$ so $\Delta(f, f'') = \Delta(f, f') + \Delta(f', f'') \geq \Delta(f, f')$.[1]

Finally, for intuition, in Theorem 7 we specialise Theorem 5 to the linear setting and compute the bound exactly. Assumptions and technical conditions are given in Section 2 along with an outline of the ideas of Elesedy and Zaidi [8] on which we build. Related works are discussed in Section 5.

## 2 Background and Preliminaries

In this section we provide a brief introduction to reproducing kernel Hilbert spaces (RKHS) and the ideas we borrow from Elesedy and Zaidi [8]. Throughout this paper, $\mathcal{H}$ with be an RKHS with kernel $k$. In Section 2.2 we state some topological and measurability assumptions that are needed for our proofs. These conditions are benign and the reader not interested in technicalities need take from Section 2.2 only that $\mu$ is $\mathcal{G}$-invariant and that the kernel $k$ is bounded and satisfies Eq. (1). We defer proofs to the Supplementary Material.

---

[1]To be completely clear: if, for instance, it so happens that projecting the RKHS onto a space of invariant predictors before doing KRR gives lower test risk than projecting afterwards (what we are calling feature averaging), then our result applies in that case too.

## 2.1 RKHS Basics

A Hilbert space is an inner product space that is complete with respect to the norm topology induced by the inner product. A reproducing kernel Hilbert space (RKHS) $\mathcal{H}$ is Hilbert space of real functions $f : \mathcal{X} \to \mathbb{R}$ on which the evaluation functional $\delta_x : \mathcal{H} \to \mathbb{R}$ with $\delta_x[f] = f(x)$ is continuous $\forall x \in \mathcal{X}$, or, equivalently is a bounded operator. The Riesz Representation Theorem tells us that there is a unique function $k_x \in \mathcal{H}$ such that $\delta_x[f] = \langle k_x, f \rangle_{\mathcal{H}}$ for any $f \in \mathcal{H}$, where $\langle \cdot, \cdot \rangle_{\mathcal{H}} : \mathcal{H} \times \mathcal{H} \to \mathbb{R}$ is the inner product on $\mathcal{H}$. We identify the function $k : \mathcal{X} \times \mathcal{X} \to \mathbb{R}$ with $k(x, y) = \langle k_x, k_y \rangle_{\mathcal{H}}$ as the *reproducing kernel* of $\mathcal{H}$. Using the inner product representation, one can see that $k$ is positive-definite and symmetric. Conversely, the Moore-Aronszajn Theorem shows that for any positive-definite and symmetric function $k$, there is a unique RKHS with reproducing kernel $k$. In addition, any Hilbert space admitting a reproducing kernel is an RKHS. Finally, another characterisation of $\mathcal{H}$ is as the completion of the set of linear combinations of the form $f_c(x) = \sum_{i=1}^{n} c_i k(x, x_i)$ for $c_1, \dots, c_n \in \mathbb{R}$ and $x_1, \dots, x_n \in \mathcal{X}$. For (many) more details, see [30, Chapter 4].

## 2.2 Technical Setup and Assumptions

**Input Space, Group and Measure**  Let $\mathcal{G}$ be a compact[2], second countable, Hausdorff topological group with Haar measure $\lambda$ (see [12, Theorem 2.27]). Let $\mathcal{X}$ be a non-empty Polish space admitting a finite, $\mathcal{G}$-invariant Borel measure $\mu$, with supp $\mu = \mathcal{X}$. We normalise $\mu(\mathcal{X}) = \lambda(\mathcal{G}) = 1$, the latter is possible because $\lambda$ is a Radon measure. We assume that $\mathcal{G}$ has a measurable action on $\mathcal{X}$ that we will write as $gx$ for $g \in \mathcal{G}$, $x \in \mathcal{X}$. A measurable action is one such that the map $g : \mathcal{G} \times \mathcal{X} \to \mathcal{X}$ is $(\lambda \otimes \mu)$-measurable. A function $f : \mathcal{X} \to \mathbb{R}$ is $\mathcal{G}$-invariant if $f(gx) = f(x) \ \forall x \in \mathcal{X} \ \forall g \in \mathcal{G}$. Similarly, a measure $\mu$ on $\mathcal{X}$ is $\mathcal{G}$-invariant if $\forall g \in \mathcal{G}$ and any $\mu$-measurable $B \subset \mathcal{X}$ the pushforward of $\mu$ by the action of $\mathcal{G}$ equals $\mu$, i.e. $(g_* \mu)(B) = \mu(B)$. This means that if $X \sim \mu$ then $gX \sim \mu$ $\forall g \in \mathcal{G}$. We will make use of the fact that the Haar measure is $\mathcal{G}$-invariant when $\mathcal{G}$ acts on itself by either left or right multiplication, the latter holding because $\mathcal{G}$ is compact. Up to normalisation, $\lambda$ is the unique measure on $\mathcal{G}$ with this property.

**The Kernel and the RKHS**  Let $k : \mathcal{X} \times \mathcal{X} \to \mathbb{R}$ be a measurable kernel with RKHS $\mathcal{H}$ such that $k(\cdot, x) : \mathcal{X} \to \mathbb{R}$ is continuous for any $x \in \mathcal{X}$. Assume that $\sup_{x \in \mathcal{X}} k(x, x) = M_k < \infty$ and note that this implies that $k$ is bounded since

$$k(x, x') = \langle k_x, k_{x'} \rangle_{\mathcal{H}} \leq \|k_x\|_{\mathcal{H}} \|k_{x'}\|_{\mathcal{H}} = \sqrt{k(x, x)} \sqrt{k(x', x')} \leq M_k.$$

Every $f \in \mathcal{H}$ is $\mu$-measurable, bounded and continuous by [30, Lemmas 4.24 and 4.28] and in addition $\mathcal{H}$ is separable using [30, Lemma 4.33]. These conditions allow the application of [30, Theorem 4.26] to relate $\mathcal{H}$ to $L_2(\mathcal{X}, \mu)$ in the proofs building towards Lemma 3, given in the Supplementary Material. We assume that the kernel satisfies, for all $x, y \in \mathcal{X}$,

$$\int_{\mathcal{G}} k(gx, y) \, \mathrm{d}\lambda(g) = \int_{\mathcal{G}} k(x, gy) \, \mathrm{d}\lambda(g). \tag{1}$$

Equation (1) is our main assumption and we will make frequent use of it. For Eq. (1) to hold, it is sufficient to have $k(gx, y)$ equal to $k(x, gy)$ or $k(x, g^{-1}y)$, where the latter uses compactness (hence unimodularity) of $\mathcal{G}$ to change variables $g \leftrightarrow g^{-1}$. Highlighting two special cases: any inner product kernel $k(x, x') = \kappa(\langle x, x' \rangle)$ such that the action of $\mathcal{G}$ is unitary with respect to $\langle \cdot, \cdot \rangle$ satisfies Eq. (1), as does any stationary kernel $k(x, x') = \kappa(\|x - x'\|)$ with norm that is preserved by $\mathcal{G}$ in the sense that $\|gx - gx'\| = \|x - x'\|$ for any $g \in \mathcal{G}$, $x, x' \in \mathcal{X}$. If the norm/inner product is Euclidean, then any orthogonal representation of $\mathcal{G}$ will have this property.[3]

## 2.3 Invariance from a Function Space Perspective

Given a function $f : \mathcal{X} \to \mathbb{R}$ we can define a corresponding orbit-averaged function $\mathcal{O}f : \mathcal{X} \to \mathbb{R}$ with values

$$\mathcal{O}f(x) = \int_{\mathcal{G}} f(gx) \, \mathrm{d}\lambda(g).$$

---

[2]The set of compact groups covers almost all invariances in machine learning, including all finite groups (such as permutations or reflections), many continuous groups such as rotations or translations on a bounded domain (e.g. an image) and combinations thereof.

[3]An orthogonal representation of $\mathcal{G}$ on $\mathbb{R}^d$ is an action of $\mathcal{G}$ via orthogonal matrices, i.e. a homomorpishm $\mathcal{G} \to O(d)$.

$\mathcal{O}f$ will exist whenever $f$ is $\mu$-measurable. Note that $\mathcal{O}$ is a linear operator and, from the invariance of $\lambda$, $\mathcal{O}f$ is always $\mathcal{G}$-invariant. Interestingly, $f$ is $\mathcal{G}$-invariant *only* if $f = \mathcal{O}f$. Elesedy and Zaidi [8] use these observations to characterise invariant functions and study their generalisation properties. In short, this work extends these insights to kernel methods. Along the way, we will make frequent use of the following (well known) facts about $\mathcal{O}$.

**Lemma 1** ([8, Propositions 24 and 25]). *A function $f$ is $\mathcal{G}$-invariant if and only if $\mathcal{O}f = f$. This implies that $\mathcal{O}$ is a projection operator, so can have only two eigenvalues $0$ and $1$.*

**Lemma 2** ([8, Lemma 1]). *$\mathcal{O} : L_2(\mathcal{X}, \mu) \to L_2(\mathcal{X}, \mu)$ is well-defined and self-adjoint. Hence, $L_2(\mathcal{X}, \mu)$ has the orthogonal decomposition*

$$L_2(\mathcal{X}, \mu) = S \oplus A$$

*where $S = \{f \in L_2(\mathcal{X}, \mu) : f \text{ is } \mathcal{G} \text{ invariant}\}$ and $A = \{f \in L_2(\mathcal{X}, \mu) : \mathcal{O}f = 0\}$.*

The meaning of Lemma 2 is that any $f \in L_2(\mathcal{X}, \mu)$ has a (unique) decomposition $f = \bar{f} + f^\perp$ where $\bar{f} = \mathcal{O}f$ is $\mathcal{G}$-invariant and $\mathcal{O}f^\perp = 0$. A noteworthy consequence of this setup, as discussed in [8], is a provably non-negative generalisation benefit for feature averaging. In particular, for any predictor $f \in L_2(\mathcal{X}, \mu)$, if the target $f^* \in L_2(\mathcal{X}, \mu)$ is $\mathcal{G}$-invariant then the test error $R(f) = \mathbb{E}_{X \sim \mu}[(f(X) - f^*(X))^2]$ satisfies

$$R(f) - R(\bar{f}) = \|f^\perp\|^2_{L_2(\mathcal{X}, \mu)} \geq 0.$$

The same holds if the target is corrupted by independent, zero mean (additive) noise. [4]

## 3 Induced Structure of $\mathcal{H}$

In this section we present Lemma 3, which is an analog of Lemma 2 for RKHSs. Lemma 3 shows that for any compact group $\mathcal{G}$ and RKHS $\mathcal{H}$, if the kernel for $\mathcal{H}$ satisfies the assumptions in Section 2.2, then $\mathcal{H}$ can be viewed as being built from two orthogonal RKHSs, one consisting of invariant functions and another of those that vanish when averaged over $\mathcal{G}$. Later in the paper, this decomposition will allow us to analyse the generalisation benefit of invariant predictors.

It may seem at first glance that Lemma 3 should follow immediately from Lemma 2, but this is not the case. First, it is not obvious that for any $f \in \mathcal{H}$, its orbit averaged version $\mathcal{O}f$ is also in $\mathcal{H}$. Moreover, in contrast with $L_2(\mathcal{X}, \mu)$, an explicit form for the inner product on $\mathcal{H}$ is not immediate, which means that some work is needed to check that $\mathcal{O}$ is self-adjoint on $\mathcal{H}$. These are important requirements for the proofs of both Lemmas 2 and 3 and we establish them, along with $\mathcal{O}$ being continuous on $\mathcal{H}$, in the Supplementary Material. The assumption that the kernel satisfies Eq. (1) plays a central role.

**Lemma 3.** *$\mathcal{H}$ admits the orthogonal decomposition*

$$\mathcal{H} = \overline{\mathcal{H}} \oplus \mathcal{H}_\perp$$

*where $\overline{\mathcal{H}} = \{f \in \mathcal{H} : f \text{ is } \mathcal{G}\text{-invariant}\}$ and $\mathcal{H}_\perp = \{f \in \mathcal{H} : \mathcal{O}f = 0\}$. Moreover, $\overline{\mathcal{H}}$ is an RKHS with kernel*

$$\bar{k}(x, y) = \int_{\mathcal{G}} k(x, gy) \, d\lambda(g)$$

*and $\mathcal{H}_\perp$ is an RKHS with kernel*

$$k^\perp(x, y) = k(x, y) - \bar{k}(x, y).$$

*Finally, $\bar{k}$ is $\mathcal{G}$-invariant in both arguments.*

As stated earlier, the perspective provided by Lemma 3 will support our analysis of generalisation. Just as with Lemma 2, Lemma 3 says that any $f \in \mathcal{H}$ can be written as $f = \bar{f} + f^\perp$ where $\bar{f}$ is $\mathcal{G}$-invariant and $\mathcal{O}f^\perp = 0$ with $\langle \bar{f}, f^\perp \rangle_{\mathcal{H}} = 0$. As an aside, $\bar{k}$ happens to qualify as a *Haar Integration Kernel*, a concept introduced by Haasdonk, Vossen, and Burkhardt [10]. We will see that a notion of effective dimension of the RKHS $\mathcal{H}_\perp$ with kernel $k^\perp$ governs the generalisation gap between an arbitrary predictor $f$ and its invariant version $\mathcal{O}f$. This effective dimension arises from the spectral theory of an integral operator related to $k$, which we develop in the next section.

---

[4]The result [8, Lemma 1] is given for equivariance, of which invariance is a special case.

### 3.1 Spectral Representation and Effective Dimension

In this section we consider the spectrum of an integral operator related to the kernel $k$. This analysis will ultimately allow us to define a notion of effective dimension of $\mathcal{H}_\perp$ that we will later see is important to the generalisation of invariant predictors. While the integral operator setup is standard, the use of this technique to identify an effective dimension of $\mathcal{H}_\perp$ is novel.

Define the integral operator $S_k : L_2(\mathcal{X}, \mu) \to \mathcal{H}$ by

$$S_k f(x) = \int_\mathcal{X} k(x, x') f(x') \, \mathrm{d}\mu(x').$$

One way of viewing things is that $S_k$ assigns to every element in $L_2(\mathcal{X}, \mu)$ a function in $\mathcal{H}$. On the other hand, every $f \in \mathcal{H}$ is bounded so has $\|f\|_{L_2(\mathcal{X},\mu)} < \infty$ and belongs to some element of $L_2(\mathcal{X}, \mu)$. We write $\iota : \mathcal{H} \to L_2(\mathcal{X}, \mu)$ for the *inclusion map* that sends $f$ to the element of $L_2(\mathcal{X}, \mu)$ that contains $f$. In the Supplementary Material we show that $\iota$ is injective, so any element of $L_2(\mathcal{X}, \mu)$ contains at most one $f \in \mathcal{H}$.

One can define $T_k : L_2(\mathcal{X}, \mu) \to L_2(\mathcal{X}, \mu)$ by $T_k = \iota \circ S_k$, and [30, Theorem 4.27] says that $T_k$ is compact, positive, self-adjoint and trace-class. In addition, $L_2(\mathcal{X}, \mu)$ is separable by [7, Proposition 3.4.5], because $\mathcal{X}$ is Polish and $\mu$ is a Borel measure, so has a countable orthonormal basis. Hence, by the Spectral Theorem, there exists a countable orthonormal basis $\{\tilde{e}_i\}$ for $L_2(\mathcal{X}, \mu)$ such that $T_k \tilde{e}_i = \lambda_i \tilde{e}_i$ where $\lambda_1 \geq \lambda_2 \geq \cdots \geq 0$ are the eigenvalues of $T_k$. Moreover, since $\iota$ is injective, for each of the $\tilde{e}_i$ for which $\lambda_i > 0$ there is a unique $e_i \in \mathcal{H}$ such that $\iota e_i = \tilde{e}_i$ and $S_k \tilde{e}_i = \lambda_i e_i$.

Now, since $\iota k_x \in L_2(\mathcal{X}, \mu)$ we have

$$\iota k_x = \sum_i \langle \iota k_x, \tilde{e}_i \rangle_{L_2(\mathcal{X},\mu)} \tilde{e}_i = \sum_i (S_k \tilde{e}_i)(x) \tilde{e}_i = \sum_i \lambda_i e_i(x) \tilde{e}_i. \tag{2}$$

From now on we permit ourself to drop the $\iota$ to reduce clutter. We use the above to define

$$j(x, y) = \langle k_x, k_y \rangle_{L_2(\mathcal{X},\mu)}, \quad \bar{j}(x, y) = \langle \bar{k}_x, \bar{k}_y \rangle_{L_2(\mathcal{X},\mu)} \quad \text{and} \quad j^\perp(x, y) = \langle k_x^\perp, k_y^\perp \rangle_{L_2(\mathcal{X},\mu)}.$$

These quantities will appear again in our analysis of the generalisation of invariant kernel methods. Indeed, we will see later in this section that $\mathbb{E}[j^\perp(X, X)]$ is a type of effective dimension of $\mathcal{H}_\perp$. Following Eq. (2), one finds the series representations given below in Lemma 4.

The reader may have noticed that our setup is very similar to the one provided by Mercer's theorem. However, we do not assume compactness of $\mathcal{X}$ and so the classical form of Mercer's Theorem does not apply. This aspect of our work is a feature, rather than a bug: the loosening of the compactness condition allows application to common settings such as $\mathcal{X} = \mathbb{R}^n$. For generalisations of Mercer's Theorem see [31] and references therein.

**Lemma 4.** We have
$$j = \bar{j} + j^\perp.$$
Furthermore, let $\bar{e}_i = \mathcal{O}e_i$ and $e_i^\perp = e_i - \bar{e}_i$ then

$$j(x, y) = \sum_i \lambda_i^2 e_i(x) e_i(y), \quad \bar{j}(x, y) = \sum_i \lambda_i^2 \bar{e}_i(x) \bar{e}_i(y), \quad \text{and} \quad j^\perp(x, y) = \sum_i \lambda_i^2 e_i^\perp(x) e_i^\perp(y).$$

Finally, the function $\sum_i \lambda_i^2 \bar{e}_i \otimes e_i^\perp : \mathcal{X} \times \mathcal{X} \to \mathbb{R}$ with values $(x, y) \mapsto \sum_i \lambda_i^2 \bar{e}_i(x) e_i^\perp(y)$ vanishes everywhere.

Before turning to generalisation, we describe how the above quantities can be used to define a measure effective dimension. We define
$$\dim_{\text{eff}}(\mathcal{H}) = \mathbb{E}[j(X, X)]$$
where $X \sim \mu$. Applying Fubini's theorem, we find

$$\dim_{\text{eff}}(\mathcal{H}) = \sum_i \lambda_i^2 \, \mathbb{E}[e_i(X)^2] = \sum_i \lambda_i^2 \|\tilde{e}_i\|_{L_2(\mathcal{X},\mu)}^2 = \sum_i \lambda_i^2. \tag{3}$$

The series converges by the comparison test because $\lambda_i \geq 0$ and $\sum_i \lambda_i = \text{Tr}(T_k) < \infty$ (using Lidskii's theorem) because $T_k$ is trace-class. We have $\dim_{\text{eff}}(\mathcal{H}) = \text{Tr}(T_k^2)$ and we can think of this

(very informally) as taking $L_2(\mathcal{X}, \mu)$, pushing it through $\mathcal{H}$ twice using $T_k$ and then measuring its size. Now because $j = \bar{j} + j^\perp$ we get

$$\dim_{\text{eff}}(\mathcal{H}) = \dim_{\text{eff}}(\overline{\mathcal{H}}) + \dim_{\text{eff}}(\mathcal{H}_\perp)$$

with

$$\dim_{\text{eff}}(\mathcal{H}_\perp) = \sum_i \lambda_i^2 \|\tilde{e}_i^\perp\|_{L_2(\mathcal{X}, \mu)}^2 = \text{Tr}(T_k^2) - \text{Tr}((\mathcal{O}T_k)^2)$$

where $\tilde{e}_i^\perp = \iota e_i^\perp$. Again, very informally, this can be thought of as pushing $L_2(\mathcal{X}, \mu)$ through $\mathcal{H}_\perp$ twice and measuring the size of the output. In the next section we will consider the generalisation of kernel ridge regression and find that $\dim_{\text{eff}}(\mathcal{H}_\perp)$ plays a critical role.

# 4 Generalisation

In this section we apply the theory developed in Section 3 to study the impact of invariance on kernel ridge regression with an invariant target. We analyse the generalisation benefit of feature averaging, finding a strict benefit when the target is $\mathcal{G}$-invariant.

## 4.1 Kernel Ridge Regression

Given input/output pairs $\{(x_i, y_i) : i = 1, \ldots, n\}$ where $x_i \in \mathcal{X}$ and $y_i \in \mathbb{R}$, kernel ridge regression (KRR) returns a predictor that solves the optimisation problem

$$\underset{f \in \mathcal{H}}{\operatorname{argmin}} \, C(f) \quad \text{where} \quad C(f) = \sum_{i=1}^n (f(x_i) - y_i)^2 + \rho\|f\|_\mathcal{H}^2 \tag{4}$$

and $\rho > 0$ is the regularisation parameter. KRR can be thought of as performing ridge regression in a possibly infinite dimensional feature space $\mathcal{H}$. The representer theorem tells us that the solution to this problem is of the form $f(x) = \sum_{i=1}^n \alpha_i k_{x_i}(x)$ where $\alpha \in \mathbb{R}^n$ solves

$$\underset{\alpha \in \mathbb{R}^n}{\operatorname{argmin}} \left\{ \|\boldsymbol{Y} - K\alpha\|_2^2 + \rho\alpha^\top K\alpha \right\}, \tag{5}$$

$\boldsymbol{Y} \in \mathbb{R}^n$ is the standard row-stacking of the training outputs with $\boldsymbol{Y}_i = y_i$ and $K$ is the kernel Gram matrix with $K_{ij} = k(x_i, x_j)$. We consider solutions of the form[5] $\alpha = (K + \rho I)^{-1}\boldsymbol{Y}$ which results in the predictor

$$f(x) = k_x(\boldsymbol{X})^\top (K + \rho I)^{-1}\boldsymbol{Y}$$

where $k_x(\boldsymbol{X}) \in \mathbb{R}^n$ is the vector with components $k_x(\boldsymbol{X})_i = k_x(x_i)$. We will compare the generalisation performance of this predictor with that of its averaged version

$$\bar{f} = \bar{k}_x(\boldsymbol{X})^\top (K + \rho I)^{-1}\boldsymbol{Y} \in \overline{\mathcal{H}}.$$

To do this we look at the generalisation gap.

## 4.2 Generalisation Gap

The generalisation gap is a quantity that compares the expected test performances of two predictors on a given task. Given a probability distribution $\mathbb{P}$, data $(X, Y) \sim \mathbb{P}$ and loss function $l$ defining a supervised learning task, we define the generalisation gap between two predictors $f$ and $f'$ to be

$$\Delta(f, f') = \mathbb{E}[l(f(X), Y)] - \mathbb{E}[l(f'(X), Y)]$$

where the expectations are conditional on the given realisations of $f, f'$ if the predictors are random. In this paper we consider $l(a, b) = (a - b)^2$ the squared-error loss and we will assume $Y = f^*(X) + \xi$ for some target function $f^*$ where $\xi$ is has mean 0, finite variance and is independent of $X$. In this case, the generalisation gap reduces to

$$\Delta(f, f') = \mathbb{E}[(f(X) - f^*(X))^2] - \mathbb{E}[(f'(X) - f^*(X))^2].$$

Clearly, if $\Delta(f, f') > 0$ then we expect strictly better test performance from $f'$ than $f$.

---

[5]When $K$ is a positive definite matrix this will be the *only* solution. If $K$ is singular then $\exists c \in \mathbb{R}^n$ with $\sum_{ij} K_{ij}c_ic_j = \|\sum_i c_i k_{x_i}\|_\mathcal{H}^2 = 0$ so $\sum_i c_i k_{x_i}$ is identically 0 and $\forall f \in \mathcal{H}$ we get $\sum_i c_i f(x_i) = 0$ (see [18, Section 4.6.2]). Clearly, this can't happen if $\mathcal{H}$ is sufficiently expressive. In any case, the chosen $\alpha$ is the minimum in Euclidean norm of all possible solutions.

### 4.3 Generalisation Benefit of Feature Averaging

We are now in a position to give our main result, which is a characterisation of the generalisation benefit of invariance in kernel methods. This is in some sense a generalisation of [8, Theorem 6] and we will return to this comparison later. We emphasise that Theorem 5 holds under quite general conditions that cover many practical applications.

**Theorem 5.** Let the training data be $\{(X_i, Y_i) : i = 1, \ldots, n\}$ i.i.d. with $Y_i = f^*(X_i) + \xi_i$ where $X_i \sim \mu$, $f^* \in L_2(\mathcal{X}, \mu)$ is $\mathcal{G}$-invariant and bounded, and $\{\xi_i : i = 1, \ldots, n\}$ are independent of each other and the $\{X_i\}$, with $\mathbb{E}[\xi_i] = 0$ and $\mathbb{E}[\xi_i^2] = \sigma^2 < \infty$. Let $f = \operatorname{argmin}_{f \in \mathcal{H}} C(f)$ be the solution to Eq. (4) and let $\bar{f} = \mathcal{O}f \in \overline{\mathcal{H}}$ be the result of applying feature averaging to $f$, then the generalisation gap with the squared-error loss satisfies

$$\mathbb{E}[\Delta(f, \bar{f})] \geq \frac{\mathbb{E}[f^*(X)^2 j^\perp(X, X)] + \sigma^2 \dim_{\text{eff}}(\mathcal{H}_\perp)}{(\sqrt{n}M_k + \rho/\sqrt{n})^2}$$

where each term is non-negative and

$$\dim_{\text{eff}}(\mathcal{H}_\perp) := \operatorname{Tr}(T_k^2) - \operatorname{Tr}((\mathcal{O}T_k)^2) = \mathbb{E}[j^\perp(X, X)] = \sum_\alpha \lambda_\alpha^2 \|\tilde{e}_\alpha^\perp\|_{L_2(\mathcal{X}, \mu)}^2 \geq 0$$

is the *effective dimension* of $\mathcal{H}_\perp$.

Theorem 5 shows that feature averaging is provably beneficial in terms of generalisation if the mean of the target distribution is invariant. If $\mathcal{H}$ contains any functions that are not $\mathcal{G}$-invariant then the lower bound is strictly positive. One might think that, given enough training examples, the solution $f$ to Eq. (4) would *learn* to be $\mathcal{G}$-invariant. Theorem 5 shows that this cannot happen unless the number of examples dominates the effective dimension of $\mathcal{H}_\perp$.

Recall the subspace $A$ in Lemma 2. The role of $\dim_{\text{eff}}(\mathcal{H}_\perp)$ mirrors that of $\dim A$ in [8, Theorem 6] and in the context of the theorem (linear models) $A$ can be thought of as $\mathcal{H}_\perp$ when $k$ is the linear kernel. In this sense Theorem 5 is a generalisation of [8, Theorem 6]. It is for this reason that we believe that, although the constant $M_k$ in the denominator is likely not optimal, the $O(1/n)$ rate that matches [8] is tight. We leave a more precise analysis of the constants to future work.

The second term in the numerator can be interpreted as quantifying the differences in bias. One has by the definition of $j^\perp$, that

$$\mathbb{E}[f^*(X)^2 j^\perp(X, X)] = \int_\mathcal{X} f^*(y)^2 k^\perp(x, y)^2 \, \mathrm{d}\mu(x) \, \mathrm{d}\mu(y) \tag{6}$$

using $j^\perp(x, y) = \int_\mathcal{X} k^\perp(t, x) k^\perp(t, y) \, \mathrm{d}\mu(t)$. We also have the following proposition.

**Proposition 6.**

$$\int_\mathcal{X} f^*(y)^2 k^\perp(x, y)^2 \, \mathrm{d}\mu(x) \, \mathrm{d}\mu(y) = \int_\mathcal{X} f^*(y)^2 \left(k(x, y)^2 - \bar{k}(x, y)^2\right) \mathrm{d}\mu(x) \, \mathrm{d}\mu(y)$$

For intuition, we present a simple special case of Theorem 5. In particular, the next result shows that Eq. (6) reduces to an approximation error that is reminiscent of the one in [8, Theorem 6] in a linear setting. For the rest of this section we find it helpful to refer to the action $\phi$ of $\mathcal{G}$ explicitly, writing $\phi(g)x$ instead of $gx$.

**Theorem 7.** Assume the setting and notation of Theorem 5. In addition, let $\mathcal{X} = \mathbb{S}_{d-1}$ be the unit $d - 1$ sphere and let $\mu = \operatorname{Unif}(\mathcal{X})$. Let $\mathcal{G}$ act via an orthogonal representation $\phi$ on $\mathcal{X}$ and define the matrix $\Phi = \int_\mathcal{G} \phi(g) \, \mathrm{d}\lambda(g)$. Let $k(x, y) = x^\top y$ be the linear kernel and suppose $f^*(x) = \theta^\top x$ for some $\theta \in \mathbb{R}^d$. Then the bound in Theorem 5 becomes

$$\mathbb{E}[\Delta(f, \bar{f})] \geq \frac{1}{(\sqrt{n} + \rho/\sqrt{n})^2} \left(\frac{d - \|\Phi\|_F^2}{d^2} + \frac{(d - \|\Phi\|_F^2)\|\theta\|_2^2}{d^2(d + 2)}\right)$$

where $\|\cdot\|_F$ is the Frobenius norm. The first term in the parentheses is exactly $\dim_{\text{eff}}(\mathcal{H}_\perp)$ and the second term is exactly $\mathbb{E}[f^*(X)^2 j^\perp(X, X)]$.

One can confirm that the generalisation gap cannot be negative in Theorem 7 using Jensen's inequality

$$\|\Phi\|_{\mathrm{F}}^2 = \left\|\int_{\mathcal{G}} \phi(g)\,\mathrm{d}\lambda(g)\right\|_{\mathrm{F}}^2 \leq \int_{\mathcal{G}} \|\phi(g)\|_{\mathrm{F}}^2\,\mathrm{d}\lambda(g) = \int_{\mathcal{G}} \mathrm{Tr}(\phi(g)^\top \phi(g))\,\mathrm{d}\lambda(g) = \mathrm{Tr}(I) = d$$

because the representation $\phi$ is orthgonal.

The matrix $\Phi$ in Theorem 7 can be computed analytically for various $\mathcal{G}$ and in the linear setting describes the importance of the symmetry to the task. For instance, in the simple case that $\mathcal{G} = S_d$ the permutation group on $d$ elements and $\phi$ is the natural representation in terms of permutation matrices, we have $\Phi = \frac{1}{d}\mathbf{1}\mathbf{1}^\top$ where $\mathbf{1} \in \mathbb{R}^d$ is the vector of all 1s. In this case, since the target is assumed to be $\mathcal{G}$-invariant, we must have $\theta = t\mathbf{1}$ for some $t \in \mathbb{R}$. Specifically, Theorem 7 then asserts

$$\mathbb{E}[\Delta(f, \bar{f})] \geq \frac{(d-1)(dt^2 + d + 2)}{d^2(d+2)(\sqrt{n} + \rho/\sqrt{n})^2}.$$

## 5   Related Work

Incorporating invariance into machine learning models is not a new idea. The majority of modern applications concern neural networks, but previous works have used kernels [10, 22], support vector machines [25] and polynomial feature spaces [26, 27]. Indeed, early work also considered invariant neural networks [33], using methods that seem to have been rediscovered in [23]. Modern implementations include invariant/equivariant convolutional architectures [4, 6] that are inspired by concepts from mathematical physics and harmonic analysis [13, 5]. Some of these models even enjoy universal approximation properties [19, 35].

The earliest attempt at theoretical justification for invariance of which we are aware is [1], which roughly states that enforcing invariance cannot increase the VC dimension of a model. Anselmi et al. [2] and Mroueh, Voinea, and Poggio [21] propose heuristic arguments for improved sample complexity of invariant models. Sokolic et al. [28] build on the work of Xu and Mannor [34] to obtain a generalisation bound for certain types of classifiers that are invariant to a finite set of transformations, while Sannai and Imaizumi [24] obtain a bound for models that are invariant to finite permutation groups. The PAC Bayes formulation is considered in [16, 17].

The above works guarantee only a worst-case improvement and it was not until very recently that Elesedy and Zaidi [8] derived a strict benefit for invariant/equivariant models. Our work is similar to [8] in that we provide a provably strict benefit, but differs in its application to kernels and RKHSs as opposed to linear models. We are careful to state that our setting does not directly reduce to that of [8, Theorem 6] for two reasons. First, [8, Theorem 6] considers $\mathcal{G}$ invariant linear models without regularisation. This may turn out to be accessible by a $\rho \to 0^+$ limit (the so called ridgeless limit) of Theorem 5. More importantly, linear regression is equivalent to kernel regression with the linear kernel. However, the linear kernel can be unbounded (e.g. on $\mathbb{R}$), so does not meet our technical conditions in Section 2.2. We conjecture that the boundedness assumption on $k$ can be removed, or at least with mild care weakened to hold $\mu$-almost-surely.

Also very recently, Mei, Misiakiewicz, and Montanari [20] analyse the generalisation benefit of invariance in kernels and random feature models. Our results differ from [20] in some key aspects. First, Mei, Misiakiewicz, and Montanari [20] focus on kernel ridge regression with an invariant inner product kernel whereas we study symmetrised predictors from more general kernels. Second, they obtain an expression for the generalisation error that is conditional on the training data and in terms of the projection of the predictor onto a space of high degree polynomials, while we are able to integrate against the training data and express the generalisation benefit directly in terms of properties of the kernel and the group.

## 6   Discussion

We have demonstrated a provably strict generalisation benefit for feature averaging in kernel ridge regression. In doing this we have leveraged an observation on the structure of RKHSs under the action of compact groups. We believe that this observation is applicable to other kernel methods too.

There are many possibilities for future work. As we remarked in the introduction, there is an established connection between kernels and wide neural networks via the neural tangent kernel. Using this connection, generalisation properties of wide, invariant neural networks might be accessible

through the techniques of this paper. Another natural extension of this paper is to equivariant (sometimes called *steerable*) matrix valued kernels. Approximate invariance may be handled by adding an approximation term to the bound in our main result. Finally, the ideas of this paper should also be applicable to Gaussian processes.

## Acknowledgments and Disclosure of Funding

We thank Sheheryar Zaidi for many helpful discussions in the early stages of this project and Yee Whye Teh for suggesting the application of [8] to kernels. Additionally, we thank Varun Kanade and Yee Whye Teh for advice and support throughout this and other projects. This work was supported in part by the UK EPSRC CDT in Autonomous Intelligent Machines and Systems (grant reference EP/L015897/1).

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
