*Proof.* From Lemma 1 we know that $\mathcal{O}$ is a projection operator. Since it is self-adjoint, $\mathcal{O}$ is even an orthogonal projection on $\mathcal{H}$: let $h_S$ have eigenvalue $1$ and $h_A$ have eigenvalue $0$ under $\mathcal{O}$, then

$$\langle h_S, h_A \rangle_{\mathcal{H}} = \langle \mathcal{O}h_S, h_A \rangle_{\mathcal{H}} = \langle h_S, \mathcal{O}h_A \rangle_{\mathcal{H}} = 0.$$

Therefore, by linearity, for any $f \in \mathcal{H}$ we can write $f = \bar{f} + f^\perp$ where $\bar{f} = \mathcal{O}f \in \overline{\mathcal{H}}$ is $\mathcal{G}$-invariant and $f^\perp = f - \mathcal{O}f \in \mathcal{H}_\perp$ and these terms are mutually orthogonal.

---

[4] The result [8, Lemma 1] is given for equivariance, of which invariance is a special case.

By the linearity of $\mathcal{O}$, it is clear that $\overline{\mathcal{H}} = \mathcal{O}\mathcal{H}$ is an inner product space. It is easy to show that $\mathcal{O}$ being continuous implies $\overline{\mathcal{H}}$ is complete. Thus $\overline{\mathcal{H}}$ is a Hilbert space, and an RKHS since the evaluation functional is clearly continuous on $\overline{\mathcal{H}} \subset \mathcal{H}$. For any $h_S \in \overline{\mathcal{H}}$ we have

$$h_S(x) = \langle h_S, k_x \rangle_{\mathcal{H}} = \langle h_S, \mathcal{O}k_x \rangle_{\mathcal{H}} = \langle h_S, \bar{k}_x \rangle_{\mathcal{H}}$$

and the uniqueness afforded by the Riesz representation theorem tells us that the reproducing kernel for $\overline{\mathcal{H}}$ is $\bar{k}(x,y) = \int_{\mathcal{G}} k(x, gy) \, d\lambda(g)$. We have $\| \mathrm{id} - \mathcal{O} \| \leq 2$ and we can do the same argument to show that $\mathcal{H}_\perp$ is an RKHS with reproducing kernel $k^\perp$ as claimed. Note that one can write $k^\perp(x,y) = \langle k_x^\perp, k_y^\perp \rangle_{\mathcal{H}}$ so it must be positive-definite. The $\mathcal{G}$-invariance of $\bar{k}(x,y)$ in both arguments is immediate from Eq. (1) and Lemma 1. $\qquad \square$

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

*Proof.* We show in Lemma C.2 that $\mathcal{O}$ and $S_k$ commute on $L_2(\mathcal{X}, \mu)$ and $\mathcal{O}$ is self-adjoint on $L_2(\mathcal{X}, \mu)$ by Lemma 1, so $\mathcal{O}$ and $\iota$ (the adjoint of $S_k$ by [32, Theorem 4.26]) must also commute. The first comment is then immediate from the observation that if $a \in \overline{\mathcal{H}}$ and $b \in \mathcal{H}_\perp$ one has

$$\langle \iota a, \iota b \rangle_{L_2(\mathcal{X},\mu)} = \langle \iota \mathcal{O}a, \iota b \rangle_{L_2(\mathcal{X},\mu)} = \langle \mathcal{O}\iota a, \iota b \rangle_{L_2(\mathcal{X},\mu)} = \langle \iota a, \iota \mathcal{O}b \rangle_{L_2(\mathcal{X},\mu)} = 0.$$

We also have both of

$$\langle \iota \bar{k}_x, \tilde{e}_i \rangle_{L_2(\mathcal{X},\mu)} = \langle \iota k_x, \mathcal{O}\tilde{e}_i \rangle_{L_2(\mathcal{X},\mu)} = S_k \mathcal{O}\tilde{e}_i = \mathcal{O}S_k \tilde{e}_i = \lambda_i \bar{e}_i$$

and

$$\langle \iota k_x^\perp, \tilde{e}_i \rangle_{L_2(\mathcal{X},\mu)} = \langle \iota k_x, (\mathrm{id} - \mathcal{O})\tilde{e}_i \rangle_{L_2(\mathcal{X},\mu)} = S_k(\mathrm{id} - \mathcal{O})\tilde{e}_i = (\mathrm{id} - \mathcal{O})S_k \tilde{e}_i = \lambda_i e_i^\perp.$$

Therefore $\iota \bar{k}_x = \sum_i \lambda_i \bar{e}_i(x)\tilde{e}_i$ and $\iota k_x^\perp = \sum_i \lambda_i e_i^\perp(x)\tilde{e}_i$. Taking inner products on $L_2(\mathcal{X}, \mu)$ gives the remaining results. $\square$

Before turning to generalisation, we describe how the above quantities can be used to define a measure effective dimension. We define

$$\dim_{\mathrm{eff}}(\mathcal{H}) = \mathbb{E}[j(X,X)]$$

where $X \sim \mu$. Applying Fubini's theorem, we find

$$\dim_{\mathrm{eff}}(\mathcal{H}) = \sum_i \lambda_i^2 \mathbb{E}[e_i(X)^2] = \sum_i \lambda_i^2 \|\tilde{e}_i\|_{L_2(\mathcal{X},\mu)}^2 = \sum_i \lambda_i^2. \tag{3}$$

The series converges by the comparison test because $\lambda_i \geq 0$ and $\sum_i \lambda_i = \mathrm{Tr}(T_k) < \infty$ (using Lidskii's theorem) because $T_k$ is trace-class. We have $\dim_{\mathrm{eff}}(\mathcal{H}) = \mathrm{Tr}(T_k^2)$ and we can think of this (very informally) as taking $L_2(\mathcal{X}, \mu)$, pushing it through $\mathcal{H}$ twice using $T_k$ and then measuring its size. Now because $j = \bar{j} + j^\perp$ we get

$$\dim_{\mathrm{eff}}(\mathcal{H}) = \dim_{\mathrm{eff}}(\overline{\mathcal{H}}) + \dim_{\mathrm{eff}}(\mathcal{H}_\perp)$$

with

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

*Proof.* Let $J^\perp$ be the Gram matrix with components $J_{ij}^\perp = j^\perp(X_i, X_j)$ let $u \in \mathbb{R}^n$ have components $u_i = f^*(X_i)$. We can use Lemma 2 to get

$$\Delta(f, \bar{f}) = \mathbb{E}[(k_X^\perp(\boldsymbol{X})^\top (K + \rho I)^{-1} \boldsymbol{Y})^2 | \boldsymbol{X}, \boldsymbol{Y}]$$

where $k_x^\perp(\boldsymbol{X}) \in \mathbb{R}^n$ with $k_x^\perp(\boldsymbol{X})_i = k_x^\perp(X_i)$. Let $\boldsymbol{\xi} \in \mathbb{R}^n$ have components $\boldsymbol{\xi}_i = \xi_i$ then one finds

$$\mathbb{E}[\Delta(f, \bar{f}) | \boldsymbol{X}] = \mathbb{E}[(k_X^\perp(\boldsymbol{X})^\top (K + \rho I)^{-1} u)^2 | \boldsymbol{X}] + \mathbb{E}[(k_X^\perp(\boldsymbol{X})^\top (K + \rho I)^{-1} \boldsymbol{\xi})^2 | \boldsymbol{X}]$$
$$= u^\top (K + \rho I)^{-1} J^\perp (K + \rho I)^{-1} u + \sigma^2 \operatorname{Tr}\left(J^\perp (K + \rho I)^{-2}\right)$$

where the first equality follows because $\boldsymbol{\xi}$ has mean 0 and the second comes from the trace trick. Consider the first term. We have

$$u^\top (K + \rho I)^{-1} J^\perp (K + \rho I)^{-1} u = \operatorname{Tr}((K + \rho I)^{-1} J^\perp (K + \rho I)^{-1} u u^\top),$$

applying Corollary B.2 twice and using Lemma B.3 with boundedness of the kernel gives

$$u^\top (K + \rho I)^{-1} J^\perp (K + \rho I)^{-1} u \geq \lambda_{\min}((K + \rho I)^{-1})^2 \operatorname{Tr}(J^\perp u u^\top) \geq \frac{u^\top J^\perp u}{(M_k n + \rho)^2}$$

so

$$\mathbb{E}[u^\top (K + \rho I)^{-1} J^\perp (K + \rho I)^{-1} u] \geq \frac{\mathbb{E}[u^\top J^\perp u]}{(M_k n + \rho)^2} = \frac{\sum_{ij} \mathbb{E}[f^*(X_i) f^*(X_j) j^\perp(X_i, X_j)]}{(M_k n + \rho)^2}.$$

For the first term, it remains to show that the above vanishes when $i \neq j$.

**Claim.** $\mathbb{E}[f^*(X_i) f^*(X_j) j^\perp(X_i, X_j)] = 0$ when $i \neq j$.

*Proof of claim.* Using Lemma 4 we have

$$\mathbb{E}[f^*(X_i) f^*(X_j) j^\perp(X_i, X_j)] = \mathbb{E}\left[\sum_\alpha \lambda_\alpha^2 f^*(X_i) f^*(X_j) e_\alpha^\perp(X_i) e_\alpha^\perp(X_j)\right].$$

Define

$$F_N(X_i, X_j) = \sum_{\alpha=1}^N \lambda_\alpha^2 f^*(X_i) f^*(X_j) e_\alpha^\perp(X_i) e_\alpha^\perp(X_j)$$

then clearly $F_N(X_i, X_j) \to F(X_i, X_j)$ as $N \to \infty$ where

$$F(X_i, X_j) = \sum_\alpha \lambda_\alpha^2 f^*(X_i) f^*(X_j) e_\alpha^\perp(X_i) e_\alpha^\perp(X_j).$$

On the other hand, since $i \neq j$, the mean of each term is just

$$\mathbb{E}[f^*(X) e_\alpha^\perp(X)]^2 = \langle \iota f^*, \tilde{e}_\alpha^\perp \rangle_{L_2(\mathcal{X}, \mu)}^2 = 0$$

by the $\mathcal{G}$-invariance of $f^*$ and the orthogonality in Lemma 2. It follows by linearity of expectation that $\mathbb{E}[F_N(X_i, X_j)] = 0$ for all $N \geq 0$. Now, both $f^*$ and $e_\alpha^\perp$ are bounded so there's a constant $B$ such that

$$|F_N(X_i, X_j)| \leq B \sum_{\alpha=1}^N \lambda_\alpha^2 \quad \text{and} \quad |F(X_i, X_j)| \leq B \sum_\alpha \lambda_\alpha^2$$

and the final sum is finite following the comments after Eq. (3). We can therefore apply Lebesgue's dominated convergence theorem [13, Theorem 1.21] to get that

$$0 = \mathbb{E}[F_N(X_i, X_j)] \overset{N \to \infty}{\longrightarrow} \mathbb{E}[F(X_i, X_j)]] = 0$$

as required. $\qquad\square$

Moving to the second term, we have again by two applications of Corollary B.2 and then Lemma B.3 with boundedness of the kernel that

$$\operatorname{Tr}\left(J^{\perp}(K+\rho I)^{-2}\right) \geq \lambda_{\min}\left((K+\rho I)^{-2}\right) \operatorname{Tr}(J^{\perp}) \geq \frac{\operatorname{Tr}(J^{\perp})}{(M_k n + \rho)^2}$$

and then

$$
\begin{aligned}
\frac{1}{n}\mathbb{E}[\operatorname{Tr}(J^{\perp})] &= \frac{1}{n}\sum_{i=1}^{n}\mathbb{E}\left[\sum_{\alpha}\lambda_{\alpha}^2 e_{\alpha}^{\perp}(X_i)e_{\alpha}^{\perp}(X_i)\right] \\
&= \sum_{\alpha}\lambda_{\alpha}^2 \|\tilde{e}_{\alpha}^{\perp}\|_{L_2(\mathcal{X},\mu)}^2 \\
&= \sum_{\alpha}\lambda_{\alpha}^2 - \sum_{\alpha}\lambda_{\alpha}^2 \|\mathcal{O}\tilde{e}_{\alpha}\|_{L_2(\mathcal{X},\mu)}^2 \\
&= \operatorname{Tr}(T_k^2) - \operatorname{Tr}(T_k^2\mathcal{O})
\end{aligned}
$$

where we exchange the expectation and sum using Fubini's theorem. Considering the sum in the second line, note that $\|\tilde{e}_{\alpha}\|_{L_2(\mathcal{X},\mu)}^2 = 1 = \|\mathcal{O}\tilde{e}_{\alpha}\|_{L_2(\mathcal{X},\mu)}^2 + \|\tilde{e}_{\alpha}^{\perp}\|_{L_2(\mathcal{X},\mu)}^2$ by Lemma 2 so the sum converges if $\sum_i \lambda_i^2$ converges, which we establised in the comment after Eq. (3). $\qquad\square$

Theorem 5 shows that feature averaging is provably beneficial in terms of generalisation if the mean of the target distribution is invariant. If $\mathcal{H}$ contains any functions that are not $\mathcal{G}$-invariant then the lower bound is strictly positive. One might think that, given enough training examples, the solution $f$ to Eq. (4) would *learn* to be $\mathcal{G}$-invariant. Theorem 5 shows that this cannot happen unless the number of examples dominates the effective dimension of $\mathcal{H}_{\perp}$.

Recall the subspace $A$ in Lemma 2. The role of $\dim_{\mathrm{eff}}(\mathcal{H}_{\perp})$ mirrors that of $\dim A$ in [8, Theorem 6] and in the context of the theorem (linear models) $A$ can be thought of as $\mathcal{H}_{\perp}$ when $k$ is the linear kernel. In this sense Theorem 5 is a generalisation of [8, Theorem 6]. It is for this reason that we believe that, although the constant $M_k$ in the denominator is likely not optimal, the $O(1/n)$ rate that matches [8] is tight. We leave a more precise analysis of the constants to future work.

The second term in the numerator can be interpreted as quantifying the differences in bias. One has by the definition of $j^{\perp}$, that

$$\mathbb{E}[f^*(X)^2 j^{\perp}(X,X)] = \int_{\mathcal{X}} f^*(y)^2 k^{\perp}(x,y)^2 \,\mathrm{d}\mu(x)\,\mathrm{d}\mu(y) \qquad (6)$$

using $j^{\perp}(x,y) = \int_{\mathcal{X}} k^{\perp}(t,x)k^{\perp}(t,y)\,\mathrm{d}\mu(t)$. We also have the following proposition.

**Proposition 6.**

$$\int_{\mathcal{X}} f^*(y)^2 k^{\perp}(x,y)^2\,\mathrm{d}\mu(x)\,\mathrm{d}\mu(y) = \int_{\mathcal{X}} f^*(y)^2\left(k(x,y)^2 - \bar{k}(x,y)^2\right)\mathrm{d}\mu(x)\,\mathrm{d}\mu(y)$$

*Proof.* Using $k^{\perp} = k - \bar{k}$

$$
\begin{aligned}
\int_{\mathcal{X}} f^*(y)^2 k^{\perp}(x,y)^2\,\mathrm{d}\mu(x)\,\mathrm{d}\mu(y) = &\int_{\mathcal{X}} f^*(y)^2 k(x,y)^2\,\mathrm{d}\mu(x)\,\mathrm{d}\mu(y) \\
&- 2\int_{\mathcal{X}} f^*(y)^2 \bar{k}(x,y)k(x,y)\,\mathrm{d}\mu(x)\,\mathrm{d}\mu(y) \\
&+ \int_{\mathcal{X}} f^*(y)^2 \bar{k}(x,y)^2\,\mathrm{d}\mu(x)\,\mathrm{d}\mu(y)
\end{aligned}
$$

while, since $f^*$ is $\mathcal{G}$-invariant, $\mu$ is $\mathcal{G}$-invariant (by assumption) and $\mathcal{G}$ is unimodular (because it is compact),

$$
\begin{aligned}
\int_{\mathcal{X}} f^*(y)^2 \bar{k}(x,y)k(x,y)\,\mathrm{d}\mu(x)\,\mathrm{d}\mu(y) &= \int_{\mathcal{X}}\int_{\mathcal{G}} f^*(gy)^2\,\mathrm{d}\lambda(g)\bar{k}(x,y)k(x,y)\,\mathrm{d}\mu(x)\,\mathrm{d}\mu(y) \\
&= \int_{\mathcal{X}} f^*(y)^2 \int_{\mathcal{G}} \bar{k}(x,gy)k(x,gy)\,\mathrm{d}\lambda(g)\,\mathrm{d}\mu(x)\,\mathrm{d}\mu(y) \\
&= \int_{\mathcal{X}} f^*(y)^2 \bar{k}(x,y)^2\,\mathrm{d}\mu(x)\,\mathrm{d}\mu(y)
\end{aligned}
$$

where the final line follows because $\bar{k}$ is $\mathcal{G}$-invariant. $\qquad\square$

For intuition, we present a simple special case of Theorem 5. In particular, the next result shows that Eq. (6) reduces to an approximation error that is reminiscent of the one in [8, Theorem 6] in a linear setting. For the rest of this section we find it helpful to refer to the action $\phi$ of $\mathcal{G}$ explicitly, writing $\phi(g)x$ instead of $gx$.

**Theorem 7.** Assume the setting and notation of Theorem 5. In addition, let $\mathcal{X} = \mathbb{S}_{d-1}$ be the unit $d-1$ sphere and let $\mu = \mathrm{Unif}(\mathcal{X})$. Let $\mathcal{G}$ act via an orthogonal representation $\phi$ on $\mathcal{X}$ and define the matrix $\Phi = \int_{\mathcal{G}} \phi(g)\,\mathrm{d}\lambda(g)$. Let $k(x,y) = x^\top y$ be the linear kernel and suppose $f^*(x) = \theta^\top x$ for some $\theta \in \mathbb{R}^d$. Then the bound in Theorem 5 becomes

$$\mathbb{E}[\Delta(f, \bar{f})] \geq \frac{1}{(\sqrt{n} + \rho/\sqrt{n})^2} \left( \frac{d - \|\Phi\|_F^2}{d^2} + \frac{(d - \|\Phi\|_F^2)\|\theta\|_2^2}{d^2(d+2)} \right)$$

where $\|\cdot\|_F$ is the Frobenius norm. The first term in the parentheses is exactly $\dim_{\mathrm{eff}}(\mathcal{H}_\perp)$ and the second term is exactly $\mathbb{E}[f^*(X)^2 j^\perp(X,X)]$.

*Proof.* We will make use of the Einstein convention of summing repeated indices. Since $\mu$ is finite, by Fubini's theorem [13, Theorem 1.27] we are free to integrate in any order throughout the proof. First of all notice that $\sup_x k(x,x) = 1$ so $M_k = 1$. Now observe that

$$\bar{k}(x,y) = x^\top \int_{\mathcal{G}} \phi(g)y\,\mathrm{d}\lambda(g) = x^\top \Phi y.$$

Then the first term in the numerator becomes

$$
\begin{aligned}
\dim_{\mathrm{eff}}(\mathcal{H}_\perp) &= \mathbb{E}[j^\perp(X,X)] \\
&= \mathbb{E}[j(X,X)] - \mathbb{E}[\bar{j}(X,X)] \\
&= \int_X k(x,y)^2\,\mathrm{d}\mu(x)\,\mathrm{d}\mu(y) - \int_X \bar{k}(x,y)^2\,\mathrm{d}\mu(x)\,\mathrm{d}\mu(y) \\
&= \int_{\mathcal{X}} x_a x_b y_a y_b\,\mathrm{d}\mu(x)\,\mathrm{d}\mu(y) - \int_X x_a x_b y_c y_e \Phi_{ac} \Phi_{be}\,\mathrm{d}\mu(x)\,\mathrm{d}\mu(y) \\
&= \frac{1}{d} - \frac{1}{d^2}\|\Phi\|_F^2.
\end{aligned}
$$

Where $x_a$ is the $a^{\mathrm{th}}$ component of $x$, and so on. Now for the second term. We calculate each term of the right hand side of Proposition 6 separately. We know that

$$f^*(x)^2 k(x,y)^2 = (\theta^\top x)^2 (x^\top y)^2 = \theta_a \theta_b y_c y_e x_a x_b x_c x_e.$$

Integrating $y$ first, we get

$$
\begin{aligned}
\int_{\mathcal{X}} f^*(x)^2 k(x,y)^2\,\mathrm{d}\mu(x)\,\mathrm{d}\mu(y) &= \int_{\mathcal{X}} \theta_a \theta_b y_c y_e x_a x_b x_c x_e\,\mathrm{d}\mu(x)\,\mathrm{d}\mu(y) \\
&= \frac{1}{d}\int_{\mathcal{X}} \theta_a \theta_b x_a x_b\,\mathrm{d}\mu(x) \\
&= \frac{1}{d^2}\|\theta\|_2^2
\end{aligned}
$$

Similarly, we find

$$
\begin{aligned}
\int_{\mathcal{X}} f^*(x)^2 \bar{k}(x,y)^2\,\mathrm{d}\mu(x)\,\mathrm{d}\mu(y) &= \int_{\mathcal{X}} \theta_a \theta_b x_a x_b x_c x_e y_f y_h \Phi_{cf} \Phi_{eh}\,\mathrm{d}\mu(x)\,\mathrm{d}\mu(y) \\
&= \frac{1}{d}\theta_a \theta_b \Phi_{cf} \Phi_{ef} \int_{\mathcal{X}} x_a x_b x_c x_e\,\mathrm{d}\mu(x).
\end{aligned}
$$

The 4-tensor $\int_{\mathcal{X}} x_a x_b x_c x_e\,\mathrm{d}\mu(x)$ is isotropic, so must have the form

$$\int_{\mathcal{X}} x_a x_b x_c x_e\,\mathrm{d}\mu(x) = \alpha \delta_{ab}\delta_{ce} + \beta \delta_{ac}\delta_{be} + \gamma \delta_{ae}\delta_{bc}$$

(see, e.g. Hodge [11]). By symmetry and exchangeability we have $\alpha = \beta = \gamma$. Then contracting the first two indices gives

$$\int_{\mathcal{X}} x_a x_a x_c x_e \, \mathrm{d}\mu(x) = \frac{1}{d}\delta_{ce} = \alpha(d+2)\delta_{ce}$$

so $\alpha = \frac{1}{d(d+2)}$ and we end up with

$$\int_{\mathcal{X}} f^*(x)^2 \bar{k}(x,y)^2 \, \mathrm{d}\mu(x) \, \mathrm{d}\mu(y) = \frac{\|\theta\|_2^2 \|\Phi\|_F^2 + 2\|\Phi\theta\|_2^2}{d^2(d+2)} = \frac{\|\theta\|_2^2(\|\Phi\|_F^2 + 2)}{d^2(d+2)}$$

where the second equality comes from

$$\theta^\top \Phi x = \int_{\mathcal{G}} \theta^\top \phi(g) x \, \mathrm{d}\lambda(g) = \int_{\mathcal{G}} f^*(\phi(g)x) \, \mathrm{d}\lambda(g) = f^*(x) = \theta^\top x$$

for any $x \in \mathcal{X}$. Putting everything together gives the result. $\qquad\square$

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

## A  Notation and Definitions

Trace of a linear operator $A : V \to V$ on a inner product space $V$ is defined by

$$\mathrm{Tr}(A) = \sum_i \langle Av_i, v_i \rangle$$

where the collection $\{v_i\}$ forms an orthonormal basis of $V$. In this paper we will only encounter situations in which the basis is countable. This expressions is independent of the basis. We say $A$ is *trace-class* if $\mathrm{Tr}(A) < \infty$.

For any matrix $A \in \mathbb{R}^{n \times n}$, we define $\|A\|_2 = \sup_{x \in \mathbb{R}^n} \frac{\|Ax\|_2}{\|x\|}$ which is the operator norm induced by the Euclidean norm. For any symmetric matrix $A$, we denote by $\lambda_{\max}(A)$ and $\lambda_{\min}(A)$ the largest and smallest eigenvalues of $A$ respectively.

## B  Useful Results

This section contains some results that are relied upon elsewhere in the paper.

**Lemma B.1** (Mori [22])**.** Let $A, B \in \mathbb{R}^{n \times n}$ and suppose $B$ is symmetric. Define $A' = \frac{1}{2}(A + A^\top)$, then
$$\lambda_{\min}(A') \, \mathrm{Tr}(B) \leq \mathrm{Tr}(AB) \leq \lambda_{\max}(A') \, \mathrm{Tr}(B),$$
where $\lambda_{\min}$ and $\lambda_{\max}$ denote the smallest and largest eigenvalues respectively.

**Corollary B.2.** Let $A, B \in \mathbb{R}^{n \times n}$ and suppose $A$ is symmetric, then
$$\lambda_{\min}(A) \, \mathrm{Tr}(B) \leq \mathrm{Tr}(AB) \leq \lambda_{\max}(A) \, \mathrm{Tr}(B).$$

*Proof.* Let $B' = \frac{1}{2}(B + B^\top)$, then using Lemma B.1 we have
$$\lambda_{\min}(A) \, \mathrm{Tr}(B') \leq \mathrm{Tr}(AB') \leq \lambda_{\max}(A) \, \mathrm{Tr}(B').$$
On the other hand, $\mathrm{Tr}(B') = \mathrm{Tr}(B)$ and
$$2 \, \mathrm{Tr}(AB') = \mathrm{Tr}(AB) + \mathrm{Tr}(AB^\top) = \mathrm{Tr}(AB) + \mathrm{Tr}(BA).$$

$\square$

**Lemma B.3.** Let $A \in \mathbb{R}^{n \times n}$, then

$$\|A\|_2 \le n \max_{ij} |A_{ij}|.$$

*Proof.* Let $a_i \in \mathbb{R}^n$ be the $i^{\text{th}}$ column of $A$, then

$$\sup_{\|x\|_2=1} \|Ax\|_2 = \sup_{\|x\|_2=1} \sqrt{\sum_i (a_i^\top x)^2} \le \sup_{\|x\|_2=1} \sqrt{\sum_i \|a_i\|_2^2 \|x\|_2^2} \le \sqrt{\sum_i \|a_i\|_2^2} \le \sqrt{n^2 \max_{ij} A_{ij}^2}.$$

$\square$

# C    Results leading to Lemma 3

Recall from Section 3 the integral operator $S_k : L_2(\mathcal{X}, \mu) \to \mathcal{H}$ defined by

$$S_k f(x) = \int_{\mathcal{X}} k(x, y) f(y) \, \mathrm{d}\mu(y)$$

with adjoint $\iota : L_2(\mathcal{X}, \mu) \to \mathcal{H}$.

**Lemma C.1.** The image of $L_2(\mathcal{X}, \mu)$ under $S_k$ is dense in $\mathcal{H}$ and $\iota$ is injective.

*Proof.* By [32, Theorem 4.26] $\|f\|_{L_2(\mathcal{X}, \mu)} < \infty \ \forall f \in \mathcal{H}$ and $S_k(L_2(\mathcal{X}, \mu))$ is dense in $\mathcal{H}$ if and only if the inclusion $\iota : \mathcal{H} \to L_2(\mathcal{X}, \mu)$ is injective. Injectivity of the inclusion is equivalent to the statement that for any $f, f' \in \mathcal{H}$ the set

$$A(f, f') = \{x \in \mathcal{X} : f(x) \ne f'(x)\}$$

has $A \ne \varnothing \implies \mu(A) > 0$. Continuity implies that for any $f, f' \in \mathcal{H}$, either $f = f'$ pointwise or $A(f, f')$ contains an open set. By the support of $\mu$ this implies $\mu(A) > 0$. Thus, $\iota$ is injective.    $\square$

From [8, Proposition 22] we know that $\mathcal{O} : L_2(\mathcal{X}, \mu) \to L_2(\mathcal{X}, \mu)$ is well-defined and that $\|\mathcal{O}\| \le 1$. Let the image of $L_2(\mathcal{X}, \mu)$ under $S_k$ be $\mathcal{H}_2$, then Lemma C.1 states that $\overline{\mathcal{H}_2} = \mathcal{H}$.

**Lemma C.2.** For any $f \in L_2(\mathcal{X}, \mu)$, $\mathcal{O}S_k f = S_k \mathcal{O} f \in \mathcal{H}_2$. This implies $\mathcal{O} : \mathcal{H}_2 \to \mathcal{H}_2$ is well defined.

*Proof.* $\lambda$ is a Radon measure [13, Theorem 2.27] so is finite because $\mathcal{G}$ is compact and all $f \in \mathcal{H}$ are bounded so we can apply Fubini's theorem [13, Theorem 1.27] as follows: taking $f \in L_2(\mathcal{X}, \mu)$

$$
\begin{aligned}
S_k \mathcal{O} f(x) &= \int_{\mathcal{X}} \int_{\mathcal{G}} k(x, y) f(gy) \, \mathrm{d}\lambda(g) \, \mathrm{d}\mu(y) \\
&= \int_{\mathcal{X}} \int_{\mathcal{G}} k(x, g^{-1}y) f(y) \, \mathrm{d}\lambda(g) \, \mathrm{d}\mu(y) \quad \text{invariance of } \mu \\
&= \int_{\mathcal{X}} \int_{\mathcal{G}} k(gx, y) \, \mathrm{d}\lambda(g) f(y) \, \mathrm{d}\mu(y) \quad \text{Eq. (1) then unimodularity of } \mathcal{G} \\
&= \int_{\mathcal{G}} \int_{\mathcal{X}} k(gx, y) f(y) \, \mathrm{d}\mu(y) \, \mathrm{d}\lambda(g) \quad \text{Fubini} \\
&= \mathcal{O}S_k f(x).
\end{aligned}
$$

Briefly, some detail on the application of Fubini's theorem. Since $f$ may be negative, it is required that

$$\int_{\mathcal{G}} \int_{\mathcal{X}} |k(gx, y) f(y)| \, \mathrm{d}\mu(y) \, \mathrm{d}\lambda(g) < \infty.$$

Observe that

$$
\begin{aligned}
\int_{\mathcal{G}} \int_{\mathcal{X}} |k(gx, y) f(y)| \, \mathrm{d}\mu(y) \, \mathrm{d}\lambda(g) &= \int_{\mathcal{G}} \int_{\mathcal{X}} k(gx, y) |f(y)| \, \mathrm{d}\mu(y) \, \mathrm{d}\lambda(g) \\
&= \int_{\mathcal{G}} S_k |f| (gx) \\
&\le \lambda(\mathcal{G}) \sup_{x \in \mathcal{X}} S_k |f| (x) < \infty.
\end{aligned}
$$

The final inequality follows from finiteness of $\lambda$ and the fact that $S_k |f| \in \mathcal{H}$ so is bounded.    $\square$

**Lemma C.3.** Let $a, b \in \mathcal{H}_2$ with preimages $a', b' \in L_2(\mathcal{X}, \mu)$ such that $a = S_k a'$ and $b = S_k b'$, then

$$\langle a, b \rangle_{\mathcal{H}} = \int_{\mathcal{X}} a'(x) b'(y) k(x, y) \, \mathrm{d}\mu(x) \, \mathrm{d}\mu(y).$$

*Proof.* The inner product on $\mathcal{H}$ is a bounded linear functional, hence commutes with integration. We can thus calculate

$$
\begin{aligned}
\langle a, b \rangle_{\mathcal{H}} &= \langle \int_{\mathcal{X}} a'(x) k(x, \cdot) \, \mathrm{d}\mu(x), \int_{\mathcal{X}} b'(y) k(y, \cdot) \, \mathrm{d}\mu(y) \rangle_{\mathcal{H}} \\
&= \int_{\mathcal{X}} a'(x) b'(y) \langle k_x, k_y \rangle_{\mathcal{H}} \, \mathrm{d}\mu(x) \, \mathrm{d}\mu(y) \\
&= \int_{\mathcal{X}} a'(x) b'(y) k(x, y) \, \mathrm{d}\mu(x) \, \mathrm{d}\mu(y).
\end{aligned}
$$

$\square$

**Lemma C.4.** For any $f, h \in \mathcal{H}_2$,

$$\langle \mathcal{O} f, h \rangle_{\mathcal{H}} = \langle f, \mathcal{O} h \rangle_{\mathcal{H}}.$$

*Proof.* Let $f'$ and $h'$ be the pre-images of $f$ and $h$ respectively under $S_k$. Using Lemma C.3, Fubini's theorem [13, Theorem 1.27], the $\mathcal{G}$-invariance of $\mu$ and Eq. (1) we can calculate

$$
\begin{aligned}
\langle \mathcal{O} f, h \rangle_{\mathcal{H}} &= \int_{\mathcal{X}} \int_{\mathcal{G}} f'(gx) h'(y) k(x, y) \, \mathrm{d}\lambda(g) \, \mathrm{d}\mu(x) \, \mathrm{d}\mu(y) \\
&= \int_{\mathcal{G}} \int_{\mathcal{X}} f'(x) h'(y) k(g^{-1}x, y) \, \mathrm{d}\mu(x) \, \mathrm{d}\mu(y) \, \mathrm{d}\lambda(g) \quad \mathcal{G}\text{-invariance of } \mu \\
&= \int_{\mathcal{G}} \int_{\mathcal{X}} f'(x) h'(y) k(x, g^{-1}y) \, \mathrm{d}\mu(x) \, \mathrm{d}\mu(y) \, \mathrm{d}\lambda(g) \quad \text{Eq. (1)} \\
&= \int_{\mathcal{G}} \int_{\mathcal{X}} f'(x) h'(gy) k(x, y) \, \mathrm{d}\mu(x) \, \mathrm{d}\mu(y) \, \mathrm{d}\lambda(g) \quad \mathcal{G}\text{-invariance of } \mu \\
&= \langle f, \mathcal{O} h \rangle_{\mathcal{H}}.
\end{aligned}
$$

The justification for the application of Fubini's theorem is the same as in the proof of Lemma C.2. $\square$

**Lemma C.5.** $\mathcal{O} : \mathcal{H}_2 \to \mathcal{H}_2$ is bounded and $\|\mathcal{O}\| \leq 1$.

*Proof.* Let $f \in \mathcal{H}_2$, then using Lemmas 1 and C.4 along with Cauchy-Schwarz

$$\|\mathcal{O} f\|_{\mathcal{H}}^2 = \langle \mathcal{O} f, \mathcal{O} f \rangle_{\mathcal{H}} = \langle f, \mathcal{O} f \rangle_{\mathcal{H}} \leq \|f\|_{\mathcal{H}} \|\mathcal{O} f\|_{\mathcal{H}}.$$

$\square$

**Lemma C.6.** $f \in \mathcal{H} \implies \mathcal{O} f \in \mathcal{H}$ so $\mathcal{O} : \mathcal{H} \to \mathcal{H}$ is well defined.

*Proof.* By Lemma C.1, for any $f \in \mathcal{H}$ there is a sequence $\{f_n\} \subset \mathcal{H}_2$ converging to $f$ in $\|\cdot\|_{\mathcal{H}}$. Lemma C.2 shows that $\mathcal{O} : \mathcal{H}_2 \to \mathcal{H}_2$ is well defined, so the sequence $\{\mathcal{O} f_n\} \subset \mathcal{H}_2$. By Lemma C.5 we have $\|\mathcal{O} f_n - \mathcal{O} f_m\|_{\mathcal{H}} \leq \|f_n - f_m\|_{\mathcal{H}}$ and so $\{\mathcal{O} f_n\}$ is Cauchy. By completeness of $\mathcal{H}$, $\bar{f} := \lim_{n \to \infty} \mathcal{O} f_n \in \mathcal{H}$. Moreover, $\mathcal{O}$ bounded so is also continuous and we get $\bar{f} = \lim_{n \to \infty} \mathcal{O} f_n = \mathcal{O} \lim_{n \to \infty} f_n = \mathcal{O} f$. $\square$

**Lemma C.7.** $\mathcal{O}$ is self-adjoint with respect to the inner product on $\mathcal{H}$.

*Proof.* We will make use of the continuity of the inner product on $\mathcal{H}$. First let $h \in \mathcal{H}$, $f \in \mathcal{H}_2$. We saw from the proof of Lemma C.6 that $\exists$ sequence $\{h_n\} \subset \mathcal{H}_2$ with limit $h$ and $\{\mathcal{O} h_n\} \subset \mathcal{H}_2$ with limit $\mathcal{O} h$. Then $\langle \mathcal{O} h_n, f \rangle_{\mathcal{H}} \to \langle \mathcal{O} h, f \rangle_{\mathcal{H}}$ and simultaneously, applying Lemma C.4, $\langle \mathcal{O} h_n, f \rangle_{\mathcal{H}} = \langle h_n, \mathcal{O} f \rangle_{\mathcal{H}} \to \langle h, \mathcal{O} f \rangle_{\mathcal{H}}$ so the two limits must be equal. Then assuming instead that $f \in \mathcal{H}$ one can do the same calculation again arrive at the conclusion. $\square$

**Corollary C.8.** $\mathcal{O} : \mathcal{H} \to \mathcal{H}$ is bounded with $\|\mathcal{O}\| \leq 1$. Indeed, if $\mathcal{H}$ contains any $\mathcal{G}$-invariant functions then $\|\mathcal{O}\| = 1$ and if not then $\|\mathcal{O}\| = 0$.

*Proof.* Using Lemma C.7 we can repeat the calculation in Lemma C.4. The second claim follows from Lemma 1 and the variational representation of the operator norm. □