# OpenReview forum: "Provably Strict Generalisation Benefit for Invariance in Kernel Methods"
_NeurIPS.cc/2021/Conference — NeurIPS 2021 Poster_

### Official Review · Reviewer_j3tj · 2021-07-06

**Rating:** 6
**Confidence:** 4

**Summary:**

The paper aims at quantifying the possible benefits of incorporating invariances to kernel methods. The approach takes the hypothesis returned by kernel ridge regression and enforces invariance by orbit averaging relative to actions of a compact group, i.e., the test prediction at a point $x$ is the average of predictions over the orbit set $\\{ gx\ |\ g \in G \\}$, with $G$ denoting the group defining an invariance principle. The main contribution is a theoretical result showing that for target hypotheses which are G-invariant, i.e., $f(gx)=f(x)$ for all $g \in G$ and $x \in X$, the excess risk is strictly positive (relative to kernel ridge regression hypothesis) and the sample complexity when enforcing invariance can be much lower compared to pursuing standard kernel ridge regression learning (Theorem 5). The main idea is to observe that orbit averaging is a linear operator which defines a subspace of the original RKHS and by exploiting this orthogonal decomposition one gets a lower bound on the excess risk, similar to treatments in low-rank approximations via sub-space restricted inner products.

**Limitations And Societal Impact:**

Adequately addressed

**Main Review:**

First of all, my comments are based on the version attached as supplementary materials that contains appendix and the main part of the paper in a single file.

My first observation is that the assumption in Eq. (1) is fundamental to the result and merits a more detailed discussion than the one provided. It might also be good to have this assumption listed just as one would list a definition, lemma, etc. Namely, this restricts the class of kernels to which the main result/contribution applies and, thus, impacts to standardly used kernels should be discussed in details. Furthermore, I do not find sufficiently clear the argument about the inner product kernels. This needs to be formalized and carefully introduced. At the same time, it would be good to improve motivation at this point and give an example of a group that might induce useful invariance in some ML application.

Would it be possible to provide further clarification for Eq. (5) in line 287? It is not obvious how one obtains the kernel square on the right-hand side and double-integral over X.

Why is measure $\mu$ G-invariant in the proof of Proposition 6? If this is an implication in line 290, it needs further clarification.

Clarity should be improved at several places. For example,
- what does it mean to act via an orthogonal representation on X in line 298? It is clear to me but this has not been introduced properly.
- line 305: x_a, x_b, y_a, y_b have not been introduced prior to their use? Impossible to follow after that...

When it comes to related work, it feels that the two papers below are related to the overall goal and motivation of this work:

[1] Raj et al. (AISTATS 2017). Local Group Invariant Representations via Orbit Embeddings.

[2] Lyle et al. (2020). On the Benefits of Invariance in Neural Networks.

In summary, the work provides a bound which indicates that for a certain class of kernels there are clear benefits of enforcing invariance. The paper, however, lacks an instantiation of the problem setting and explicit examples of groups and orbit averaging in action. Related work can also be improved with a more detailed and thorough covering of prior work. The writing is mostly clear and easy to follow, apart from few lines discussed above. Overall, this might be an interesting paper that could trigger some further research along these lines.

### minor comments
- line 199: index j should be I
- line 261: M has not been defined (I am assuming it is the max over diagonal entries of K or index k has been omitted)
- Lemma B.1 and Corollary B.2 appear to be broken (identical statements)


**Time Spent Reviewing:**

6 hours

---

> ### Author Response · Authors · 2021-08-09
> **Author Response**
>
> Thank you for your time and care in reviewing our work. We welcome your suggestions for improvements.
>
> We agree to make the following changes to improve the clarity of our work.
> -  Further emphasise our assumptions on the kernel, add detail on the inner product kernels and give examples of group invariances.
> -  Add a more detailed derivation for equation 5 and define an orthogonal representation explicitly.
> - The $\mathcal{G}$-invariance of $\mu$ is an assumption made on line 94 (it is not something we deduce). We will restate this in the proof.
> - $x_a$ and $y_a$ are the components of $x$ and $y$ respectively. We will state this explicitly.
> - We will fix the typos you point out and cite the works you suggest.
>
> Details on the clarifications:
> - Comment on the inner product kernel after eq 1: By the definition of a unitary representation $\langle{}gx, x'\rangle{} = \langle{}x, g^{-1}x'\rangle{}$ and then we can substitute $g^{-1} \mapsto g$ in the integral because $\mathcal{G}$ is unimodular (since it is compact, e.g. see Corollary 2.28 of [1]).
> - Derivation for line 287: $j^\perp$ is defined on line 189 so that $j^\perp(x, y) = \int_{\mathcal{X}} k^\perp(t, y) k^\perp(t, x) \mathrm{d}t$ and so $\mathbb{E}[j^\perp(X, X)f(X)] = \int f(x) \int {k^\perp(t, x)}^2 \mathrm{d}t \mathrm{d}x$ which gives equation 5.
>
> Comments:
> - To our understanding, the statements in Lemma B.1 and Corollary B.2 are different (they have slightly different assumptions). Please can you give some further explanation to help us understand?
>
> We welcome any further comments or questions you may have.
>
> References:
>
> [1] Folland, G. B. A course in abstract harmonic analysis, volume 29. CRC press, 2016.

---

> > ### Comment · Reviewer_j3tj · 2021-08-22
> > **Author Response**
> >
> > Thank you very much for a detailed response. The suggested changes do not constitute a major revision, in my opinion, and for this reason I will keep my scores.
> >
> > The clarification for line 287 cleans up the notation compared to the supplementary version and I am happy with it. Thank you also for pointing out the difference in statements of Lemma B.1 and Corollary B.2. It is easy to miss the prime in line 377. In Corollary B.2, should not the statement require that B is symmetric?

---

> > > ### Author Response · Authors · 2021-08-25
> > > **Author Response**
> > >
> > > Can you please elaborate on why you think $B$ should be symmetric in Corollary B.2? In the final line we use the fact that, for any square matrix $M$, $\text{Tr}(M) = \text{Tr}(M^\top)$.

---

> > > > ### Comment · Reviewer_j3tj · 2021-08-25
> > > > **Corollary B.2**
> > > >
> > > > My initial impression was that it might impact the inequality in line 380, but see now that what used to be $A'$ in Lemma B.1 is now actually A which is by definition symmetric and $A'=A$. Instead of $B$ from Lemma B.1 there is now $B'$ which is symmetric by definition. Think it was confusing because I assumed $B'$ was taking the role of $A'$.

---

### Official Review · Reviewer_LcrW · 2021-07-13

**Rating:** 5
**Confidence:** 4

**Summary:**

This article extends to a kernel setting the idea of Elesedy and Zaidi to leverage a decomposition of L^2 into invariant and null-orbit subspaces. The authors claim to prove the benefit for generalisation of projecting the solution of KRR on the invariant subspace. The article mainly consists of the lemmas and proofs to reach the main theorem on generalisation bounds.

**Limitations And Societal Impact:**

This is not applicable.

**Main Review:**

This article suffers from the comparison with Elesedy and Zaidi in terms of clarity and presentation, and its writing denotes a clear sense of haste. The idea is sound, since any extra, non-invariant, term would hamper the generalization. It was already presented almost with the same terms in Elesedy and Zaidi as underlined by the authors. The proofs seem novel but several points during my reading have cast doubt on their rigor (the version reviewed did not have an appendix). The absence of any numerical experiment even though it is only KRR is damaging, but this reviewer could have disconsidered this aspect if the proofs had been soundproof and without so many ellipses. Since the main body of the article is essentially devoted to the proofs, they should be crystal clear.

Major comments:

182 The tensor product is misleading since it would define an operator over H. As it does not appear in the proof, the meaning could not be clarified and the sentence remains ungrounded.

241 If $J(K+\rho){-1} u =0$ while $J u \neq 0$ then the first inequality cannot hold. No argument was provided to avoid this situation (e.g. commutation of matrices) and, even with it, it would require more care.

244 “temporarily suspending suspicion”, this reviewer could have done as asked if the authors had not considered that $f\in L_2$ implied that $f^2\in L_2$ as suggested by the inner product they used (this makes the formula in Theorem 5 clearly inconsistent).

249 “we have established”, where?

250 Why would the essential supremum exist? https://mathworld.wolfram.com/EssentialSupremum.html See the counter example with $1/x$.

Minor comments:

67, 141 his name is Riesz

176 Steinwart&Scovel 2012 discussed the question of the extension of Mercer’s theorem and would be worth citing. The reviewer acknowledges the points made by the authors.

212 footnote 1 seems irrelevant since there is a regularisation parameter $\rho>0$.

257 “being okay” is uncanny in a mathematical text

271 since all that is required is that the functions of $H$ are integrable w.r.t. \mu, boundedness should indeed be unnecessary, even a quadratic growth could be allowed for the kernel.

292 “focus” *on*


**Time Spent Reviewing:**

Three hours

---

> ### Author Response · Authors · 2021-08-10
> **Author Response**
>
> We thank you for your helpful suggestions as to how to improve our work. We respond to your comments and give clarifications below.
>
> A refined version of Theorem 5 is given in the supplementary material which we will promote to the main paper. We believe this resolves your comments related to lines 244 and 250. Clarification is given below anyway.
>
> We have improved both the content and presentation of our work in the supplementary material by adding examples and explanation. We will apply these changes to the main paper and demote the proofs to the supplementary material. We hope that this will improve the clarity and readability of our work.
>
> Major comments:
> - 189: to our understanding the tensor product notation is standard, e.g. following [1]. Are you able to suggest an alternative notation that you think is clearer?
> - 241: We will add an explanation to show that this cannot happen. For instance, as you will have noticed, the case of commuting matrices cannot hold because $J(K+\rho I)^{-1} u=0$ would mean  $(K + \rho I)^{-1} Ju =0$ which implies $Ju =0$ because $K+\rho I$ is full rank.
> - 244: In the main paper we show on line 87 that every $f \in \mathcal{H}$ is bounded and on line 78 we say that the measure $\mu$ is finite which together means that $f^2 \in L_2(\mathcal{X}, \mu)$.
> - 249: By assumption $f^*$ is $\mathcal{G}$-invariant and by definition $e^\perp \in \mathcal{H}_A$ so they are orthogonal in $L_2(\mathcal{X}, \mu)$, meaning the expectation vanishes when $i\ne j$ as claimed. We will add this explanation to the paper.
> - 250: Reference to the essential supremum is removed in the supplementary material version.
>
> Minor comments:
> - We will correct the typos and stylistic errors you point out and cite the paper you suggest, thank you.
> - 212: This footnote is relevant. Differentiating (4) gives $K \alpha = K (K+\rho I)^{-1} \mathbf{Y}$, so the solution given (which is the one used most commonly in the literature) is unique only if $K$ is non-singular as stated.
>
> References:
>
> [1] Ingo Steinwart and Andreas Christmann. Support Vector Machines. Information science and 509 statistics. Springer, 2008. ISBN: 978-0-387-77241-7.

---

> > ### Comment · Reviewer_LcrW · 2021-08-25
> > **Answer**
> >
> > I acknowledge having read the authors' answers which I am not satisfied with
> >
> > 189:"tensor product notation" the authors did not put a reference to a specific page for the book they quote, and no, tensor notation for functions designates in the general mathematical community an operator over H. Since the notation is not introduced, nor actually expressed in the proof,-it falls within the "the remaining results." of 190 I suppose- I kindly ask for proof and to express their quantities formally, e.g. $(x,y) \mapsto \sum_i \lambda_i^2 \bar e_i(x) e^\perp_i(y)$ if this is the quantity they use.
> >
> > 241: I just gave an example of conditions that would allow to avoid such case (commuting K and J), in the absence of proof from the authors' side I do not have to trust in their result. This rebuttal phase is an opportunity to resolve these questions. "We will add an explanation to show that this cannot happen." is at best hand-waving. In general Gram matrices have a non-null kernel, so the situation does occur, that lhs can vanish whereas not the rhs.
> >
> > 244: As the kernel is bounded, $f^2$ is indeed bounded for functions in the RKHS, but $f^*$ was not assumed to be bounded or in the RKHS. The same remark holds for $e^\perp$. (that Theorem 5 in the supplement does not contain this gross errors is another matter altogether)
> >
> > 250: How does the proof change if you decide to remove one of its arguments? Then either the argument was unnecessary (which would be problematic), or other assumptions have to be given.
> >
> > Minor 212: agreed, but this is not stated in the article, so the connection with the classical solution of KRR was not stressed well-enough for the reader to understand.
> >
> > I do not understand why theorems in the main differ from the supplement.
> >
> > "We have improved both the content and presentation of our work in the supplementary material by adding examples and explanation." Again you have the opportunity to tell us about this extra content, as a reviewer, it would be ill-practice to just take the authors' word for it.
> >
> > I am inclined to lower my vote consequently if I do not have the answers I asked for.

---

> > > ### Author Response · Authors · 2021-09-03
> > > **Author Response**
> > >
> > > First of all we would like to thank the reviewer, the AC and the SAC for their patience. We have had a combination of personal and professional issues that have delayed our response to the latest comments.
> > >
> > > Our responses:
> > >
> > > 189: To our understanding, the natural interpretation of the definition given on line 182 is the mapping you write. In any case, we would be happy to add the definition you suggest.
> > >
> > > 241: There is no absence of proof and the situation you describe does not occur. While we very much welcome your comments and requests for clarification, we must assert ourselves: line 241 is correct.
> > >
> > > To begin with, the situation you suggest would be in direct contradiction of the elementary eigenvalue inequality applied (which is proven in Corollary B.2 of the supplementary material). It is not consistent to claim the existence of such matrices without disagreeing with this result. Maybe we should have stated this point explicitly earlier.
> > >
> > > In any case, one can make a simple argument. $J^\perp$ is symmetric and positive semi-definite. If it were full rank then $J^\perp(K + \rho I)^{-1} u =0 \implies u = 0 \implies J^\perp u =0$ because $(K + \rho I)^{-1}$ is positive definite, so full rank. Suppose then that $J^\perp $ is not full rank. If $J^\perp (K + \rho I)^{-1}u =0$ then $(K + \rho I)^{-1}$ must map $u$ into the null of $J^\perp$. On the other hand, since $J^\perp u \ne 0$ we know that $u$ is in the rank of $J^\perp$. Since $J^\perp$ is symmetric, these spaces are orthogonal and hence $u^\top (K+\rho I)^{-1}u =0$, which is a contradiction.
> > >
> > > 244: The error here is in the statement of the theorem, in which the target function should be assumed to be bounded (or in $\mathcal{H}$). We will correct the statement. Please note that by definition $e^\perp \in \mathcal{H}$, so it is bounded along with $\tilde{e}^\perp$ (hence your statement does _not_ apply to $e^\perp$).
> > >
> > > 250: The essential supremum is covered by 244 above.

---

### Official Review · Reviewer_ZLX8 · 2021-07-14

**Rating:** 5
**Confidence:** 2

**Summary:**

The authors prove theoretically that incorporating invariance in kernel ridge regression improves generalization, when the output is invariant to the action of a compact group.

**Limitations And Societal Impact:**

This is a purely theoretical work and the authors do not discuss any potential negative societal impact.

As regards the limitations, they are actually the assumptions needed to develop the theory.

**Main Review:**

Originality:
First of all, I am not an expert in kernel methods, so I cannot judge objectively the novelty of the paper and the related work. The authors extend the function space perspective of [8] to kernel methods. In particular, they prove a strict improvement over the generalization gap. In my opinion, this is a sufficient contribution.


Quality:
The theoretical part of the paper seems to be solid. However, I think that an experimental section is missing. Even if this is a theoretical work, I believe that at least a demonstration with synthetic data is beneficial.


Clarity:
I think that the paper is ok written, taking into account its technical nature. One recommendation is to move proofs in the appendix, and use the saved space to include some concrete examples and discussions, with the goal to make the paper more accessible to a non-expert reader. Also, I believe is better to move the Sec 4.1 in the beginning of the paper, which will enable the reader to conceptualize the practical aspect of the model.


Significance:
In my opinion, this line of theoretical works/results are important in order to understand how learning methods work. Also, I think the presented result is rather interesting. Potentially, some experiments are missing to strengthen the paper. However, as a non-expert I cannot criticise its significance thoroughly.


Comments:
1) The averaged version presented between lines 215-216 needs only $\bar{k}_x(\mathbf{X})$? We should not use the averaged kernel matrix $\bar{\mathbf{X}}$?
2) In line 225 is the relation correct? In particular, it seems that if $\Delta(f, f') >0$ then the error of $f$ is higher and $f'$ performs better.

**Time Spent Reviewing:**

6

---

> ### Author Response · Authors · 2021-08-09
> **Author Response**
>
> Thank you for your time and consideration in reviewing our work.
>
> There are already many papers that implement and test invariant models, so we do not feel that adding experiments to our work will contribute meaningfully to the literature. Our objective is to explain in theoretical terms why previous applications of invariant models have been successful.
>
> We agree with your suggestion to move proofs to the appendix and including more explanation in the main paper. The supplementary material contains additional examples and explanation which we will put in place of the proofs in the main paper. We hope that this will make the paper clearer and easier to read.
>
> Responses to your comments:
> 1. The relation given in the paper is correct. Averaging is an operation that is applied to functions, so in this case it is applied to the kernel (as opposed to the design matrix).
> 2. This is a typo, we will correct it.
>
> We welcome any further comments or questions you may have.

---

> > ### Comment · Reviewer_ZLX8 · 2021-08-31
> > **After rebuttal**
> >
> > First of all I would like to thank the authors for the responses. As a non-expert in kernel methods I will not judge the technical quality of the paper. However, I see that reviewer **LcrW** has some concerns. I think that the clarity and the structure of the paper can be improved. Also, some simple experiments could have helped. Overall, I think that some improvements should take place and for this reason will keep my score to 5.

---

### Official Review · Reviewer_Bqwx · 2021-07-16

**Rating:** 8
**Confidence:** 4

**Summary:**

This paper proposes a theoretical analysis of the generalization benefit obtained for some classes of problems which exhibit invariance with respect to a group action.
Especially, the paper focuses on kernel ridge regression in the statistical model $Y = f^*(X) + \epsilon$ and shows that using orbit-averaging on the learnt model provide strictly positive benefit whenever $f^\star$ is invariant with respect to the group action used in avering.
Some decomposition of the involved RKHSs in terms of invariance is provided, and the generalization benefit is characterized in terms of the number of samples and typical quantities related to the kernel.


**Main Review:**

-Originality:
This paper extends the analysis of orbit-averaging to kernel ridge regression, while previous works focused only on linear model. This constitutes a significant leap forward for the analysis of invariant estimators. Original techniques are developed to analyse the generalization benefit,
and the literature review is furnished and fair.

-Quality:
The submission is technically sound. The results are discussed, compared, and explained as exemplified in theorem 7 where a specific case of invariance is considered to put more interpretability to a somewhat abstract bound otherwise.
Potential research directions are evoked, either on the modeling spaces or on the considered tasks.
Overall the submission is of high quality.

A few comments:
- The bound obtained here is typical of the square loss, exploiting the properties of the generalization error specific to it. Would it be possible to extend this analysis to e.g. self concordant losses, for which typical assumptions in the kernel regime include the source and capacity conditions, both complexity measures a bit different from the effective dimension defined in the paper ?
- Assumption 1 (A1) on the kernel is illustrated on two examples: inner product with unitary action, and stationnary with norm preserving action. Is this assumption hard to work with ? In the context of working with learnable kernels, could some algorithms be put in place to verify such an assumption ?

-Clarity:
The submission is very well written and pleasant to read.

-Significance:
The results are very important as learning with invariants is of interest in many practical problems, yet few theoretical hindsights can be found.

-Overall strengh:
Very well written, sound maths.
-Overall Weakness:
The page limit seems largely overstepped.


**Time Spent Reviewing:**

5

---

> ### Author Response · Authors · 2021-08-09
> **Author Response**
>
> Thank you for your close attention in reviewing our work.
>
> In response to your questions:
> - We believe that with some additional work our techniques could be extended to losses other than the square loss. We will explore self-concordant losses, thank you for your suggestion.
> - These assumptions on the kernel can be enforced for learnable kernels. For instance, given any inner product with respect to which the action is unitary, one can learn a kernel of the form $k(x, x') = h(\langle{}x, x\rangle{})$, where $h$ is a learnable mapping.
>
> We welcome any further comments or questions you may have.

---

> > ### Comment · Reviewer_Bqwx · 2021-08-25
> > **Reviewer Response**
> >
> > Thanks for your response. I will keep my score as it is.

---

### Decision · Program_Chairs · 2021-09-28

**Decision:**

Accept (Poster)

**Comment:**

The paper aims at quantifying the possible benefits of incorporating invariances to kernel methods. The approach takes the hypothesis returned by kernel ridge regression and enforces invariance by orbit averaging relative to actions of a compact group. The main contribution is a theoretical result showing that for target hypotheses that are G-invariant, the excess risk is strictly positive (relative to kernel ridge regression hypothesis) and the sample complexity, when enforcing invariance, can be much lower compared to pursuing standard kernel ridge regression learning.

The reviewers recognize the potential impact of the result and appreciate the potential of the work for motivating further research in such direction. However, they are convinced that the paper requires improvements in terms of presentation and technical quality as they pointed out in the reviews and in the subsequent discussion. We strongly invite the authors to implement the comments of the reviewers in the final version of the paper.


**Consistency Experiment:**

NeurIPS has a long history of experimentation. In 2014, NeurIPS ran an experiment in which 10% of submissions were reviewed by two independent committees to quantify the randomness in the review process. This year, we repeated a variant of this experiment to see how the quality of the review process has changed over time.  This paper was part of the experiment and was therefore assigned to two committees (consisting of reviewers, an Area Chair, and a Senior Area Chair) that reached independent decisions.  If both committees made the same recommendation, this recommendation was followed. If a single committee recommended acceptance, the paper was accepted (with the exception of a few cases in which the other committee identified what we considered a fatal flaw, e.g., an error in a key result).

Both committees reached the same decision: **Accept (Poster)**

The other committee assigned to the paper recommended **Accept (Poster)**.  You can find the other set of reviews, along with any follow up discussion with the authors here:
https://openreview.net/forum?id=yKdYdQbo22W